# Cryobiopsy in Lung Cancer Diagnosis—A Literature Review

**DOI:** 10.3390/medicina57040393

**Published:** 2021-04-19

**Authors:** Mărioara Simon, Ioan Simon, Paul Andrei Tent, Doina Adina Todea, Antonia Haranguș

**Affiliations:** 1“Leon Daniello” Clinical Hospital of Pulmonology, 400371 Cluj-Napoca, Romania; simonmariaro@gmail.com (M.S.); doina_adina@yahoo.com (D.A.T.); antonia.harangus@yahoo.com (A.H.); 2Departament of Surgery, Iuliu Hațieganu University of Medicine and Pharmacy, 400000 Cluj-Napoca, Romania; 3Department of Oral and Maxillofacial Surgery, University of Oradea, 410087 Oradea, Romania; tent_andrei@yahoo.com; 4Departament of Pulmonology, Iuliu Hațieganu University of Medicine and Pharmacy, 400000 Cluj-Napoca, Romania; 5Research Center for Functional Genomics, Biomedicine and Translational Medicine, “Iuliu Hatieganu” University of Medicine and Pharmacy, 400337 Cluj-Napoca, Romania

**Keywords:** bronchoscopy, biopsy, cryosurgery, lung neoplasms

## Abstract

Optimizing the diagnosis of lung cancer represents a challenge, as well as a necessity, for improving the low survival of these patients. Flexible bronchoscopy with forceps biopsy is one of the key diagnostic procedures used for lung tumors. The small sample size and crush artifacts are several factors that can often limit access to a complete diagnosis, therefore leading to the need of repeating the bronchoscopy procedure or other invasive diagnostic methods. The bronchoscopic cryobiopsy is a recent technique that proved its utility in the diagnosis of both endobronchial and peripheral lung tumors. In comparison with conventional forceps biopsy, studies report a higher diagnostic yield and a superior quality of the collected samples for both the histopathological and the molecular diagnosis of lung cancer. This method shows promising results in sampling lung tissue, alone, or in conjunction with fluoroscopy or radial endobronchial ultrasound (r-EBUS). With a good safety and cost-benefit profile, this novel method has the potential to improve the diagnosis, and therefore the management of lung cancer patients. The objective of this narrative review is to provide a comprehensive review of the recent data regarding the advantages of cryobiopsy and r-EBUS in lung cancer diagnosis.

## 1. Introduction

The cryotechniques, such as cryobiopsy, cryotherapy and cryorecanalization are modern methods used for the diagnosis and treatment of pulmonary diseases. Typically, the equipment consists of three main components: a console, the cryogen and cryoprobe. The details provided by the console are the value of cryopressure, the temperature at the tip of the probe and the duration of application. The cryogens used to achieve temperatures as low as −70 °C to −89 °C, and even −196 °C, are carbon dioxide or nitrous oxide. These low temperatures can be reached in seconds. The Joule–Thompson effect stands at the center of this technology, in which a compressed gas exists with a high flow and expands quickly, creating low temperatures that lead to the adhesion of the surrounding tissue to the cryoprobe. The tissue sample is then removed during the freeze-thaw cycle. There are different cryoprobes available, rigid, semi-rigid or flexible, with multiple diameters of the tip of 1.9, 2.4 or 5.5 mm and a length between 50 cm and 90 cm [1]. The disadvantage of removing the cryoprobe en bloc with the bronchoscope has been overcome by the introduction of the disposable cryoprobes with a size of 1.1mm, which can be removed through the working channel of the bronchoscope. The disposable cryoprobes also come in sizes of 1.7 and 2.4 mm and represent a viable option, overcoming the potential loss of cooling performance and the risk of cross-contamination that comes with the re-usable cryoprobes [2].

The indications of cryobiopsy are:Biopsy of malignant endobronchial growth tumors,Tangential biopsy of malignant infiltrating tumors that are hard to sample using conventional forceps biopsy,Biopsy of benign tumors,Transbronchial biopsy in diffuse interstitial lung diseases and peripheral lung nodules [3,4].

Internationally, lung cancer remains the principal cause of cancer-related deaths in both men and women. The incidence and mortality of this disease is closely related to cigarette smoking patterns. Among the reasons for the high mortality rates are unequal access to healthcare and sociocultural barriers that can cause delayed diagnosis and treatment [5]. Lung cancer is classified into two main groups, non-small cell (NSCLC) and small cell type, with two distinct managements. Adenocarcinoma is currently the most frequent histologic subtype and the incidence increases in parallel with the incidence of lung cancer in women. Squamous cell lung cancer represents the subsequent most common subtype, histologically characterized by squamous pearl formation, intercellular bridging and keratin production. Small cell lung cancer is known for a more aggressive clinical course with extensive metastases and paraneoplastic syndromes. With the emergence of precision medicine, testing for molecular markers such as epidermal growth factor receptor (EGFR) insertions and deletions, Kirsten rat sarcoma (KRAS) mutations and anaplastic lymphoma kinase (ALK) gene rearrangements has become mandatory. In advanced stages of non-small cell lung cancer, EGFR and ALK mutations predict a better prognosis and sensitivity to target therapy [6]. Immunotherapy have revolutionized the management of NSCLC and testing for programmed death-ligand 1 (PD-L1) in these patients is highly recommended [7].

In order to improve the prognosis of lung cancer patients, a complete characterization of the tumor needs to performed, with the determination of all the molecular alterations that can be targeted by novel therapies. Therefore, using a biopsy method that provides safe and adequate lung tissue sampling, without morphological alterations, such as cryobiopsy, could provide a better choice for these patients [8]. At present, bronchoscopy is the gold-standard in diagnosing endobronchial tumors. It allows macroscopic tumor assessment and sample collection by endobronchial biopsy, lavage or bronchial brushing. Although the biopsy is collected under direct visualization, sometimes the diagnosis cannot be confirmed and the bronchoscopy needs to be repeated. Among the disadvantages of forceps biopsy are the small sample sizes and the crush artefacts that influence the quality of the histopathological analysis. The traditional approach of endobronchial lesions by flexible bronchoscopy with conventional forceps biopsy has a diagnostic yield of 72–88% [9,10]. When associating forceps biopsy with brushing, lavage and needle aspiration, the diagnostic yield increases. A correlation has been made between the diagnostic yield and the size of the biopsy sample [9,11].

Cryobiopsy could have a decisive role in improving the diagnostic of lung cancer, by providing larger and better-preserved samples for both endobronchial and peripheral tumors. The number of molecular markers that need to be determined from the biopsy sample of lung cancer patients increases constantly, therefore, focusing on adequate sampling is of utmost importance. Only a few studies have been published regarding the improved diagnosis of lung cancer by cryobiopsy, alone or in combination with other methods. There is currently no standardization regarding the adequate number of cryobiopsy samples or the time of freezing necessary for an acceptable sample. Furthermore, the concerns regarding procedure related complications in lung cancer patients, such as bleeding and pneumothorax, need to be properly addressed. To our knowledge, this is the first narrative review of literature that aims to provide a complete image of how cryobiopsy can be useful in diagnosing lung cancer (endobronchial visible by bronchoscopy or presenting as peripheral pulmonary lesions (PPLs)), with implications in further diagnostic and therapeutical decisions.

## 2. Study Selection

We performed a review by searching MEDLINE/PubMed database with the goal of identifying all articles that report cryobiopsy in patients with lung cancer. The free text database search included the following words: “cryobiopsy” and “lung cancer”. No time frame was set prior to the search. This provided 70 hits. The exclusion criteria consisted of case reports and articles that were not relevant for our topic or had an unclear methodology. Figure 1 illustrates a flowchart of the search process.

## 3. Diagnostic Yield of Cryobiopsy versus Conventional Forceps Biopsy in Endobronchial Lesions

Cryobiopsy allows endobronchial tumoral tissue sampling and extraction of the sample while still being frozen, attached to the tip of the cryoprobe. Several studies in literature attempted to make a comparison between the diagnostic yield of cryobiopsy and the conventional forceps biopsy in endobronchial lesions. A prospective, randomized, single-blinded, controlled, multicenter study by Hetzel el al. [9] included 600 patients in eight centers and defined the diagnostic yield as the number of diagnostic procedures divided by the sum between the number of diagnostic procedures and the number of nondiagnostic procedures. The diagnostic yield for endobronchial forceps biopsy was 85.1%, while for cryobiopsy it was 95.0% (*p* < 0.001), proving the superiority of cryobiopsy in lung cancer patients. A smaller, but recent study, that included 47 patients, reported a higher diagnostic yield of cryobiopsy compared to the forceps biopsy (*p* = 0.001), with no reported severe complications. Two forceps biopsies and a single cryobiopsy were performed for each patient with a randomized sequence [12].

Rubio et al. [13] reported a diagnostic yield of cryobiopsy in lung cancer of 96.77%. The authors performed a comparative analysis of forceps biopsy and cryobiopsy by sampling the same tumoral lesion in 22 patients (3 endobronchial forceps and 2 cryobiopsies with 2–3 s freezing time). The mean volume of the tissue sample was significantly larger in cryobiopsy (0.696 cm^3^ versus 0.0373 cm^3^ with forceps biopsy (*p* = 0.0014)), free of artifacts and the method was overall safe. The high diagnostic yield could be explained by the sampling technique and the deeper sedation used, that could be responsible for a more accurate targeting of the tumor. Kvale et al. reported a diagnostic yield of 74% for central tumors when using conventional forceps biopsy [14]. Aktas et al. have performed bronchoscopy with three forceps biopsies and one cryobiopsy in each of the 41 patients of their prospective study. The tumor sample was pulled with cryoprobe after a 20 s freezing time. The results have shown that cryoprobe biopsies were better than forceps biopsies (92% vs. 78%) in central tumors. Regarding the complications, there was no statistically significant difference between the two types of biopsy. Cold saline, adrenaline and argon plasma coagulation were used to resolve the bleeding. [15]

Although there are various papers concerning the diagnostic performance of cryobiopsy for endobronchial lesions, the procedure protocols differ between centers, therefore having an impact on the reported diagnostic yield. A prospective study conducted by Segmen et al. [16] on 50 patients with visible endobronchial tumors, non-diagnostic by conventional forceps biopsy, aimed to establish the optimal number of cryobiopsies needed to diagnose endobronchial malignancies. A total of four cryobiopsies were taken from 49 patients. The results reported a significant difference (*p* = 0.031) between one and two biopsies, however the third and fourth biopsies were found to be redundant.

## 4. Diagnostic Yield of Transbronchial Cryobiopsy in Peripheral Lung Cancer

Transbronchial lung biopsy (TBB) was first introduced into practice to diagnose and evaluate diffuse lung diseases [17] and it is now used as a relatively safe and noninvasive technique to acquire lung tissue in lung cancer patients. Indications for this method, that has demonstrated a high diagnostic yield, include a computed tomography (CT) image of pulmonary nodules with positive bronchus sign, lesions with “tree in bud” pattern, alveolar consolidations, reticulonodular patterns with a perilymphatic distribution [18]. The performance of TBB in these lesions can be correlated to the fact that TBB obtains tissue mainly from the centrilobular area. A downside of the TBB biopsies is that, because of their small dimension, they tend to be subject to artefacts such as crush artefacts, bubble artefacts, atelectasis, hemorrhage, telescoping of vessels, artefacts mimicking cribriform carcinoma. Image guidance by cone-beam computed tomography, fluoroscopy, electromagnetic navigation or r-EBUS can improve the diagnostic accuracy by confirming access to these peripheral lesions [19].

Transbronchial lung cryobiopsy (TBLC) for peripheral lung nodules can be performed with the use of flexible bronchoscopy and fluoroscopy. The cryoprobes used for TBB can have diameters of 1.9 or 2.4 mm. Prior to the procedure, the exact lung segment of the lesion has to be determined on the CT scan. Placing a Fogarty balloon at the proximal end of the targeted segmental bronchus is recommended. Figure 2 is schematically representing a transbronchial biopsy of a peripheral lung tumor. Once the cryoprobe is in the right position, it can be cooled for 3 to 6 s. The bronchoscope is then removed together with the cryoprobe because the biopsied tissue has a larger size than the working channel of the flexible bronchoscope and it could damage the scope during retraction. The frozen tissue biopsy is thawed in saline and subsequently fixed in formalin [20].

A retrospective study by Nasu et al. [21], comprising 53 patients, aimed to compare the effectiveness of TBLC versus TBB for peripheral lung nodules. Virtual bronchoscopic navigation was used to help trace the nodules when the target bronchus had a small dimension. After the scope was close to the target, endobronchial ultrasound with guided sheath (EBUS-GS) was inserted through the working channel and then advanced under fluoroscopic guidance. When the target was confirmed, the EBUS probe was removed and the cryoprobe was introduced on the GS. The study reported, by using univariate and multivariate analyses, that cryobiopsy with GS significantly increases diagnostic yield (OR = 11.6, *p* = 0.044) and can be used to biopsy lesions with an intratumoral air bronchogram. Imabayashi et al. [22] conducted a study using TBLC with fluoroscopic guidance, without a guided sheath, for patients with peripheral lung lesions. Before the procedure, CT data were used to prepare a virtual bronchoscopic pathway that indicated the bronchial route to the lesion. Definite access to the lesion was confirmed using radial EBUS. The authors did not use a guided sheath for TBLC in order to prevent the metal tip of the probe being blocked when passing through a strongly flexed scope, as it occurs when the lesions are situated in the apex of the lung. The final diagnostic yield of TBLC was 86.1% (31/36) with histological diagnosis. When adding stamp cytology as a diagnostic test, the diagnostic yield increased to 91.6%.

As the fluoroscopy equipment is not available in some institutions to guide the bronchoscopic examination, radial endobronchial ultrasonography (EBUS) could help in localizing the peripheral lung lesions. EBUS has proved its importance as a tool in lung cancer diagnosis [23]. The guidelines of the American College of Chest Physicians (ACCP) recommend using EBUS in order to maximize the diagnostic yield of PPLs [24]. TBNA with EBUS-guide sheath (EBUS-GS) has been recognized as a valuable strategy to diagnose PPLs with a diagnostic yield between 46% to 86.7% [25,26,27,28,29].

Schuhmann et al. [30] showed that TBLC can be used to obtain tissue from peripheral lung nodules of up to 4 cm in size. The investigators first identified the peripheral lesion by r-EBUS and compared TBLC with TBB in a randomized manner. Fluoroscopy was used only if the investigator decided it can add a benefit in locating the lesion. Each of the 31 patients received three TBB and three TBLC with a 1.2 mm cryoprobe, according to a randomized distribution of 1:1. The time of the TBLC was significantly longer when compared to TBB and the diagnostic yield was of 61.3% (19/31) for TBB and 74.2% (23/31) for TBLC, *p* = 0.42. The diagnostic yield of the lesions visualized with EBUS was 74.2%. The authors reported a very low rate of complications and concluded that a combined use of EBUS-GS and TBLC could reduce the rate of complications by providing a better detection of the lesion before sampling.

A recent prospective study [31] that evaluated the diagnostic accuracy of TBLC with EBUS-GS for PPLs supports the previous data, reporting higher diagnostic accuracy for TBLB in comparison to TBB (87% versus 82.6%). The mean volume of the TBLC sample was 0.078 cm^3^, in comparison to 0.003 cm^3^ for TBB (*p* < 0.0001). All the collected TBLC specimens were analyzed by next generation sequencing and were able to provide an adequate quality and quantity of DNA for analysis. Another prospective study by Taton et al. [32], on 32 patients, assessed the diagnostic yield of electromagnetic navigation bronchoscopy (ENB) with TBLC in comparison with TBB for lung nodules with less than 2 cm in diameter. Eight samples were collected in total, six by TBB and two by TBLC. The samples collected by TBLC were on average five times larger than those by TBB. TBLC had a higher diagnostic yield than TBB (69%, respectively 38%, *p* = 0.017). Almost half of the patients (15/32) had mild or moderate bleeding, and one patient suffered a pneumothorax. Novel quality data from a prospective, double-blind, randomized controlled trial (CYRUS), concerning TBLC with r-EBUS guidance, reported a trend towards reduced bleeding and requirement for additional diagnostic interventions when r-EBUS was used to guide TBLC. Unfortunately, the data did not reach statistical significance (*p* = 0.2059) [33]. r-EBUS with or without GS was also used in 2 retrospective studies [34,35] and 2 prospective studies [36,37]. Table 1 contains the main characteristics of the studies that describe the utility of TBLC for the diagnostic of PPLs.

Peripheral lung lesions continue to represent a diagnostic dilemma. TBLC with r-EBUS guidance is feasible and can be safely used to obtain histological samples of larger volume, with an improved diagnostic yield as compared to TBB. An extended procedure time may be acceptable, as the samples are larger and of a better quality, with fewer artifacts. This fact may reduce the number of biopsies needed.

## 5. Cryobiopsy and Molecular Tests

Almost two thirds of patients with NSCLC have an advanced stage disease at the time of diagnosis [39]. The current management of advanced NSCLC has been individualized and is determined by immunohistochemical and molecular characterization of the lung tumor. However, samples obtained by conventional biopsy from PPLs are usually small and represent a challenge for the pathologist [40]. Therefore, cryobiopsy may be a useful tool for overcoming this problem. 

Testing for epidermal growth factor receptor (EGFR) mutation lays the cornerstone for NSCLC therapy in advanced stages. EGFR is one of the most relevant genetic alteration in NSCLC, as it is targeted by tyrosine kinase inhibitors with a better side effect profile and an improved progression free survival as compared to standard chemotherapy [41]. Therefore, in order to have a precise molecular characterization of the lung tumor, adequate quantity and quality tissue samples are needed [42]. Haentschel et al. [43] attempted to evaluate the detection rate of EGFR alterations in cryobiopsy samples in comparison to samples obtained by other biopsy techniques (forceps biopsy, fine needle aspiration) in NSCLC patients. In their retrospective analysis, the authors reported that cryobiopsy may be able to increase the number of NSCLC patients in advanced stages that could receive targeted therapy, by increasing the detection rate of the mutation (21.6% for cryobiopsy group versus 13.8% for non-cryobiopsy group, *p* < 0.05). Sanger sequencing was used for EGFR mutation detection and most patients were treatment-naïve at the time of biopsy. The explanation for the false negative EGFR test results could be explained by the tumoral spatial heterogeneity and the imbalance between cancer cells and normal tissue in the sample, DNA damage and artifacts caused by formalin fixation. A larger sample, as the one obtained by cryobiopsy, provides a larger amount of material with an improved capacity to augment the malignant content [44].

For the patients with advanced NSCLC that do not present targeted molecular alterations, the recent advances in lung cancer immunotherapy could be the key to a better survival rate. The checkpoint inhibitors targeting the programmed death-1 (PD-1) pathway have revealed good clinical responses with manageable side effects [45]. A recent study by Arimura et al. [37] assessed the tumor cell count and PD-L1 expression in samples obtained by TBLC combined with EBUS-GS for PPLs and compared them with data from samples obtained by conventional TBB, in 16 patients with lung cancer. Tissue samples obtained by both TBLC and TBB were fixed with formalin, stained with hematoxylin and eosin staining and underwent immunohistochemistry analysis for histopathologic subtyping and PD-L1 expression. The cut off values for PD-L1 were classified as ≥50% and ≥ 1%. The total and average number of tumor cells was significantly larger in the TBLC sample compared to the TBB sample, on account of the higher volume of the sample. TBLC samples were considered to be more appropriate for gene mutation analysis and whole exome sequencing. In TBLC samples, for PD-L1  ≥  50%, the authors found 100% specificity, 100% positive predictive value (PPV), 100% negative predictive value (NPV) and 93.8% concordance, but for PD-L1  ≥  1%, they reported 44.4% sensitivity, 50% NPV and 56.3% concordance. They hypothesized that the results are related to the heterogeneity of PD-L1 expression and the use of different antibody. Only mild bleeding was reported in four cases.

In a large single-center, prospective, single-arm study [38] 121 patients suspected or diagnosed with primary lung tumor underwent TBLC, using flexible bronchoscopy, after having undergone TBB from the same lesion. In comparison to the TBB samples, the TBLC samples were larger in size and could make a definitive diagnosis in a larger proportion. Moreover, larger amounts of DNA and RNA could be extracted from these samples (a median of 1.60 µg DNA and 0.62 µg RNA with cryoprobe vs. 0.58 µg DNA and 0.17 µg RNA with forceps). TBLC also had the tendency to yield increased rates of PD-L1 expression >1% (51% with TBLC vs. 42% with TBB). The authors calculated a tumor mutation burden of 84 (range: 3–2396), by performing a whole-exome sequencing of 18 cryoprobe samples. In another study, the success of using next generation-sequencing for DNA analysis of TBLC specimens has also been reported. A total 17 samples were sequenced and analyzed, providing a high quality and quantity of DNA [31].

The previous presented data, even if scarce, indicates that TBLC can be a valuable tool used for obtaining tumoral samples suitable in size and quality for a complete histopathological diagnosis and mutational analysis. This technique could expand the ability to use precision medicine in order to optimize the therapy of lung cancer patients.

## 6. Advantages of Cryobiopsy

Among the most important advantages that the cryobiopsy offers, when compared to conventional forceps biopsy, is a high diagnostic yield for endobronchial malignant lesions, of up to 95% [8]. The quality of the samples collected with the cryoprobe have a greater quality because of the following:-the larger size and volume of the collected sample,-the well-preserved tissue samples for histopathologic, molecular and genetic analysis,-less crush artifacts.

The size of the sample can be determined by the operator by increasing the freezing time. The larger size of the cryoprobe samples can facilitate an accurate diagnosis, therefore enabling the possibility of target- or immuno-therapy for lung cancer patients and increasing the changes for the patients to receive the best personalized care. It can also reduce the number of additional sampling examinations needed to reach a diagnosis and the need for repeated bronchoscopies.

Given the higher morbidity and mortality of surgical biopsies, TBLC can be easily adopted as a much safer and cost-effective alternative, with a high diagnostic yield. In comparison with video-assisted thoracoscopic surgery (VATS), in diffuse lung disease, TBLC proved to have a decreased median time of hospitalization (2.6 versus 6.1 days, *p* < 0.0001) and a decreased mortality as a result of adverse events (0.3% versus 2.7%) [46].

This novel bronchoscopy method could play a major role in the management of lung diseases, particularly in lung cancer, in the near future.

## 7. Limitations and Complications of Cryobiopsy

The complications related to the surgical lung biopsy (SLB) along with the low diagnostic yield of conventional transbronchial forceps biopsy have increased the need for better biopsy methods. Unfortunately, there is a lack of uniformity in reporting the hemorrhage severity among different studies and retrospective studies can be vulnerable to reporting bias when data collection is incomplete. There are no predetermined parameters for categorizing major complications. In addition, reporting in high-volume academic centers may not be fully representative of all bronchoscopy centers.

Among the studies that described cryobiopsy for endobronchial lesions, Ehab et al. [12] reported no pneumothorax or pneumomediastinum, only mild and moderate bleeding in both procedures, with no significant statistic difference between them. Aktas et al. [15] reported 34.1% and 36.6% hemorrhage following forceps and cryoprobe biopsies, respectively (*p* > 0.05). Moderate bleeding occurred after cryobiopsy in two patients and was managed by argon plasma coagulation. Another study showed an increased risk of bleeding when more than three cryobiopsies were obtained (OR = 2.758) [16].

In the studies that have been included in this review, which described TBLC for PPLs, mild to moderate bleeding was the main procedure-related complication. These events were resolved mostly by using suction, adrenaline or cold saline instillation or inflation of endobronchial balloon [30,35]. Nasu et al. [21] reported a case where they performed a lobectomy for both controlling the bleeding and treating the cancer. One patient in this study has also developed pneumonia. Two studies used endobronchial blockers [22,36] and two reported using an endobronchial balloon, Fogarty [32] or B5-2c Olympus Medical [35]. Taton et al. [32] reported pneumothorax in one patient that required pleural drainage. The low pneumothorax rates in patients with TBLC for lung cancer might be lower than in patients with interstitial lung disease, because the fibrotic lung tissue increases the risk of procedural-related pneumothorax in these patients. Hibane et al. [33] had one event of desaturation that recovered when the procedure was temporarily stopped.

Another challenge of cryobiopsy is the relative stiffness of the probe, with one study reporting that this characteristic hindered the usage of the guide sheath when using sensitive image guidance [35]. The stiffness of the cryoprobe can influence its positioning in relation with the PPLs, often moving the tip away from the target lesion. This fact, along with the necessity for renavigation for each new cryobiopsy, prolongs the duration of the procedure.

## 8. Conclusions

Increasing diagnostic yield of bronchoscopy in lung cancer represents an important step in improving the management of this pathology and the survival rates. This narrative review confirmed that cryobiopsy can be an extremely useful tool in the diagnosis of endobronchial tumors, as well as suspect PPLs by means of TBLC. It also adds a great value in obtaining tangential samples in tumors that infiltrate the bronchi and are more difficult to sample by conventional forceps biopsy. The quality and quantity of the cryobiopsy samples, with fewer crush artifacts and larger volume, does not only optimize the histopathological diagnosis, but also allows a better possibility for complete molecular characterization of the sample. In the current era, where cancer therapy is based on precision medicine, this technique could improve the chances for the lung cancer patients.

The limitations of this paper come from the fact that most of the comprised studies are retrospective, with only two studied being designed in a randomized controlled manner. The studies may have been biased in patient selection in that they included many cases that could be approached with a conventional cryoprobe that is too rigid, or a thick GS and a therapeutic bronchoscope. Therefore, it may not be meaningful to compare the diagnostic yield of cryobiopsy and forceps biopsy in peripheral lung lesions. Further research is needed to provide a meta-analysis of the current data regarding the true diagnostic yield of cryobiopsy in central and peripheral lung masses and the true value of r-EBUS.

Lung cryobiopsy is a cost-efficient method with a lower complication and mortality rate in comparison with SLB. When guided by fluoroscopy and/or radial EBUS, an increase in diagnostic yield of peripheral tumors can be noticed, together with a decrease in complications associated with this method. In conclusion, cryobiopsy should be considered as a diagnostic approach in patients with endobronchial tumors, as well as peripheral lesions with high suspicion of lung cancer.

## Figures and Tables

**Figure 1 medicina-57-00393-f001:**
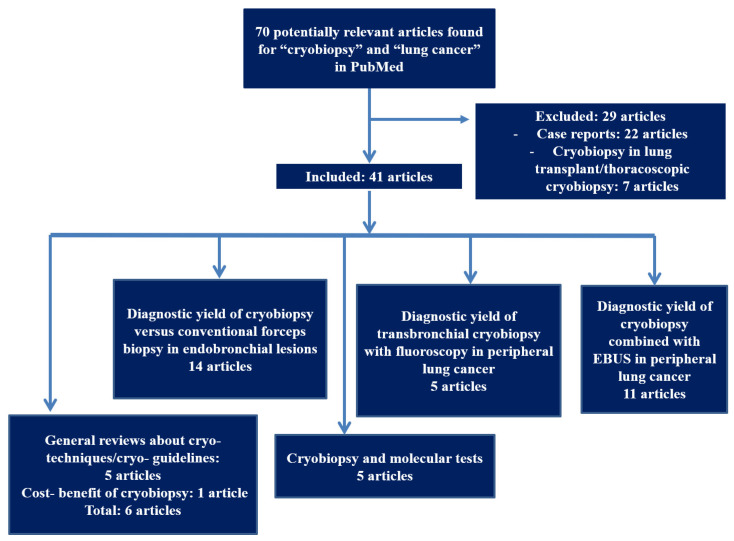
Flowchart of the search process for the review regarding cryobiopsy in lung cancer.

**Figure 2 medicina-57-00393-f002:**
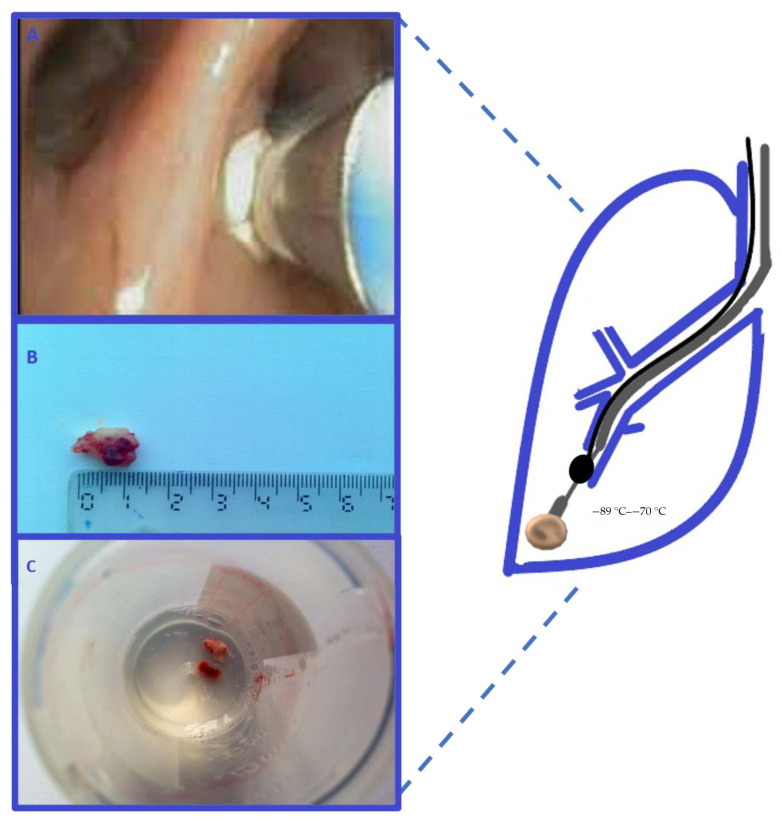
Transbronchial cryobiopsy of a peripheral lung tumor. The cryoprobe (grey) is introduced through the fiberoptic bronchoscope 6.2 mm working channel. The probe is cooled for 5–6 s (image **A**). A Fogarty balloon (black) is placed in the proximal lobar bronchus from the lesion and is inflated after the biopsy. The cryobiopsy sample has a larger size of >10 mm (images **B**,**C**).

**Table 1 medicina-57-00393-t001:** Characteristics of the studies describing the value of TBLC and r-EBUS for PPLs.

Author	Study Design	Number of Patients	Median Size of Biopsy Sample TBB vs. TBLC	Use of EBUS or Fluoroscopy	Comments
Nasu et al. [21]	Retrospective	53	2.62/14.1 mm^2^	r-EBUS + GS	Cryobiopsy with GS in PPLs with positive bronchus sign significantly increases diagnostic yield (OR = 11.6, *p* = 0.044)
Imabayashi et al. [22]	Retrospective	38	NA/12.2 mm^2^	r-EBUS	Diagnostic yield of CB increased from 86.1% to 91.6% when adding stamp cytology.
Schuhmann et al. [30]	Randomized controlled study	38	4.69/11.17 mm^2^	r-EBUS +GS	Time of TBLC was significantly longer when compared to TBB. Diagnostic yield was of 61.3% (19/31) for TBB and 74.2% (23/31) for TBLC (*p* = 0.42)
Arimura et al. [31]	Prospective	23	0.003 ± 0.0003/0.078 ± 0.008 (mean ± SEM) cm^3^	r-EBUS + GS + Fluoroscopy	Higher diagnostic accuracy for TBLB in comparison to TBB (87% versus 82.6%). Sufficient quantity and quality for DNA analysis by NGS.
Taton et. al. [32]	Prospective	32	1.1 ± 0.6/5.3 ± 0.7 mm	r-EBUS + GS	No statistically significant impact on the diagnostic yield for the location or size of nodule or the technique use (visualization with EBUS mini probe)
Hibare et al. [34]	Retrospective	55	NA	Radial EBUS ± GS ± Fluoroscopy	No significant difference was found in diagnostic yield between TBB or TBLC. 14% of lesions could not be located by r-EBUS
Kho et al. [35]	Retrospectve	114	NA	Radial EBUS ± GS ± Fluoroscopy	The addition of rapid on-site cytology (ROSE) increased the sensitivity, specificity, PPV and diagnostic accuracy
Herath et al. [36]	Prospective	6	3.4/6.4 mm	Radial EBUS + GS	The GS was trimmed by 3 cm from the distal end of the scope for a better contact.
Udagawa et al. [38]	Prospective	121	2/15 mm^2^	Radial EBUS ± GS + Fluoroscopy	Larger amounts of DNA and RNA with TBLC (a median of 1.60 µg DNA and 0.62 µg RNA with cryoprobe vs. 0.58 µg DNA and 0.17 µg RNA with forceps).

## Data Availability

Not applicable.

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
