# Peer review of "Cryobiopsy in Lung Cancer Diagnosis—A Literature Review"

_medicina, 2021, doi:10.3390/medicina57040393_

Round 1
Reviewer 1 Report
Review medicina-1170773
Cryobiopsy in lung cancer diagnosis – a literature review
Author Mărioara Simon and colleagues
This is a well written review of the currently available literature focusing on the role and potential added value of cryobiopsy in lung cancer diagnostics.
This paper is relevant, given the need for sufficient tumor samples in order to determine the ever growing range of molecular and predictive markers needed for treatment selection in modern oncological treatment.
I have a few remarks.
Page 2 line 42: the authors list the range of available cryoprobes, but seem to focus on the reusable versions only. To my knowledge, at least in my country, the most important producer of these probes (Erbe Medizin) have stopped the distribution of these reusable ones and shifted to disposable probes. This has of course a significant impact on the costs of the procedure, but also the available probes have a different sizes; sizes remain: 1,1 mm,1,7 mm and 2,4 mm.
Introduction line 50-82: this is a nice general summary but could be shortened significantly in order to focus towards the need for adequate specimen sampling.
Line 101-128: and following Yield.
For my understanding, please define the diagnostic yield in this section, and check if all these paper did use the same definition. In my experience DY can be used quite diverse: sometimes it only means that you were able to get a sample regardless of the outcome in relation to the gold standard (surgery). Sometimes is is used only to see if the the authors were able to perform a specific test (eg EGFR) but then still you do not know how this reflects to the ideal concept of yield for a patient: take one sample and get every answer. I was triggered since the differences were that big, whereas I would have expected that in visible endobronchial tumors the added value of cryo over forceps biopsy would be very limited if you take enough samples with both tools. I would expect that you may be able to take only one or two using cryo to have enough tissue, whereas with forceps’s, as we know you should take at least 5 but preferably more.
Section 4:
I congratulate the authors with this nice overview.
In general, I would like to add a remark relating to the stiffness of the cryoprobe and the size of the samples obtained.
For navigational bronchoscopy procedures of very small lesions in the periphery of the lungs 3D image guidance seems to be essential to confirm access to these solid or GGO lesions, see for example: [https://journals.lww.com/bronchology/Fulltext/2021/01000/Cone_Beam_CT_Image_Guidance_With_and_Without.12.aspx]. When using this sensitive image guidance, it becomes clearly visible that the relative stiffness of the cryoprobe has a large influence on the positioning of the probe often moving the tip away from the target lesion. What’s more, for these procedures extended working channels are used, which are similar in size as the probe, so therefore only one sample can be obtaind using the 1,9 cryoprobes.
Section 5:
Please focus this section to answer if cryobiopy is able to perform a full spectrum NGS with multigene testing, full immunohistochemistry, and all relevant predictive markers (PDL1, TMB). EGFR testing alone is not sufficient anymore these days.
Reviewer 2 Report
The paper is an honest and accurate review of the advantage and limits of cryobiopsy of lung tissue for functional and pathological examination. It confirms what was already known based on objective data extracted from an accurate selection of pertinent literature. The paper may be published after formal review by the Editorial Assistant. I would like to suggest the Authors to increase the size of the three images on the left in Fig. 2, page 5, to increase their visibility.
